# Structural Plasticity of LL-37 Indicates Elaborate Functional Adaptation Mechanisms to Bacterial Target Structures

**DOI:** 10.3390/ijms22105200

**Published:** 2021-05-14

**Authors:** Kornelius Zeth, Enea Sancho-Vaello

**Affiliations:** 1Department of Science and Environment, Roskilde University, Universitetsvej 1, 4000 Roskilde, Denmark; 2Institute of Microbiology and Infection, College of Medical and Dental Sciences, University of Birmingham, Edgbaston, Birmingham B15 2TT, UK

**Keywords:** antimicrobial peptides, LL-37, cathelicidin, structural flexibility

## Abstract

The human cathelicidin LL-37 is a multifunctional peptide of the human innate immune system. Among the various functions of LL-37, its antimicrobial activity is important in controlling the microorganisms of the human body. The target molecules of LL-37 in bacteria include membrane lipids, lipopolysaccharides (LPS), lipoteichoic acid (LTA), proteins, DNA and RNA. In this mini-review, we summarize the entity of LL-37 structural data determined over the last 15 years and specifically discuss features implicated in the interactions with lipid-like molecules. For this purpose, we discuss partial and full-length structures of LL-37 determined in the presence of membrane-mimicking detergents. This constantly growing structural database is now composed of monomers, dimers, tetramers, and fiber-like structures. The diversity of these structures underlines an unexpected plasticity and highlights the conformational and oligomeric adaptability of LL-37 necessary to target different molecular scaffolds. The recent co-crystal structures of LL-37 in complex with detergents are particularly useful to understand how these molecules mimic lipids and LPS to induce oligomerization and fibrillation. Defined detergent binding sites provide deep insights into a new class of peptide scaffolds, widening our view on the fascinating world of the LL-37 structural factotum. Together, the new structures in their evolutionary context allow for the assignment of functionally conserved residues in oligomerization and target interactions. Conserved phenylalanine and arginine residues primarily mediate those interactions with lipids and LPS. The interactions with macromolecules such as proteins or DNA remain largely unexplored and open a field for future studies aimed at structures of LL-37 complexes. These complexes will then allow for the structure-based rational design of LL-37-derived peptides with improved antibiotic properties.

## 1. Introduction

Given the threat of a post-antibiotic era where common infections and minor injuries may kill, the discovery of new antibiotic molecule classes has high medical priority [1]. Natural and designed antimicrobial peptides (AMPs) are ancient but effective weapons, which today appear as an appealing alternative to classical antibiotics, with currently 36 candidates in clinical trials [2,3]. AMPs have co-evolved with bacteria for over millions of years without a significant loss of pathogen control [4]. Compared to small molecule antibiotics, their advantage is based on their properties to target multiple chemically different but essential cellular structures [5]. Most AMPs are positively charged, and many target the essential structures: lipopolysaccharides (LPS) and lipoteichoic acid (LTA), bacterial membranes, signal transduction protein complexes, and RNA or DNA [6,7,8]. The interaction between AMPs and membranes is experimentally the most intensively studied, and here, it is assumed that their high local concentration leads to the assembly on and perforation of bacterial membranes resulting in the breakdown of the transmembrane potential. Classically, the models describing the membrane disruption mechanism have been distinguished into the detergent-like carpet, the toroidal pore (electrostatic interactions of AMPs and lipids), and the barrel-stave model (hydrophobic and electrostatic interactions of AMPs) [2,9,10]. Today it is assumed that there are intermediate and combined mechanisms that an individual AMP may follow, and this sequence of steps also depends on the particular Gram-negative or Gram-positive bacterial cell type [11,12,13]. 

The human cathelicidin LL-37 is the best-studied AMP [14]. It is composed of a highly conserved N-terminal “cathelin” domain shared with other cathelicidins and a C-terminal extension, termed LL-37, containing a peptide that is released after extracellular proteolytic treatment at the place of activity [15,16,17]. The free amphipathic peptide contains 37 residues with 16 charged residues and a total net charge of +6 [18,19,20,21]. The sequence alignment of LL-37 orthologues from vertebrates shows the conserved hydrophobic/hydrophilic pattern that maintains the amphipathic nature of this AMP when folded and also increases the folding propensity of LL-37 due to intramolecular salt bridges [20,22]. Hydrophobic residues (in particular all phenylalanines) and the Ile20-Leu28 segment of LL-37 included in the active core region (see Figure 1A) are strongly conserved while the negatively charged and the hydrophilic uncharged residues are less strongly conserved [23]. The secondary structure and the oligomeric state of LL-37 are determined by detergents/lipids, pH, salt, and divalent anions, and these factors together influence the antimicrobial activity [24,25]. Like many AMPs, LL-37 was shown to interact with negatively charged cellular targets such as LPS, LTA, and more unspecifically with negatively charged bacterial membrane lipids [6,7,8]. Furthermore, LiaX and PhoQ, proteins belonging to two-component systems, initiate membrane remodeling or LPS modifications, respectively, in response to LL-37 interactions [26,27,28,29]. Moreover, the negatively charged DNA, RNA, and polyribosomes are targets of LL-37 [30,31]. The stabilization of neutrophil-derived DNA or neutrophil extracellular traps (NETs) against bacterial nuclease degradation is one of the most recent activities assigned to LL-37 [32,33]. This multiplicity of functions is most likely based on the conformational variability and the various oligomeric states of LL-37 allowing it to adapt to variable targets.

Among its target structures, LL-37 is best studied in the context of lipid membranes and detergent micelles, both having a strong influence on the secondary structure. Interactions of membrane proteins with lipids or lipid membranes are typically mimicked by detergent molecules. Experiments in the absence of detergents show that the helicity and oligomerization state of LL-37 is concentration-dependent with an equilibrium of monomers up to hexamers depending on the experimental conditions [20,24,25]. Oligomerization of LL-37 in solution likely occurs after helix formation and is a prerequisite for membrane interactions [24]. Hydrophobic and charged residues of LL-37 can contribute to target interactions. Studies testing the electrostatic properties of LL-37 in lipid monolayers showed a strong affinity towards the anionic dipalmitoylphosphatidylglycerol (DPPG), whereas the zwitterionic dipalmitoylphosphatidylcholine (DPPC) and dipalmitoylphosphatidylethanolamine (DPPE) monolayers remained virtually unaffected [34]. Accordingly, LL-37 was also inserted efficiently into monolayers composed of the negatively charged lipid A showing a small interaction with 1,2-dioleoyl-sn-glycero-3-[phospho-rac-(1-glycerol)] (DOPG) and lipid A within the 1,2-Dioleoyl-sn-glycero-3-phosphocholine (DOPC) monolayer [35,36]. This somewhat contradicted the previous results by Oren et al., suggesting that LL-37 can bind zwitterionic phosphatidylcholine (PC) membranes as an oligomer and mixtures of negatively charged PC/ phosphatidylserine (PS) as monomers [25]. This controversy was solved when it was proposed that the LL-37 insertion into zwitterionic membranes could be concentration-dependent, suggesting that an increase in LL-37 concentration could produce the peptide insertion into DPPC monolayers as observed by Oren et al. [34]. 

Detergents such as dodecylphosphocholine (DPC), N,N-Dimethyldodecylamine N-oxide (LDAO), and n-dodecyl-β-D-maltoside (DDM) also induced the helical folding and oligomerization of LL-37 into tetramers/hexamers without a significant influence from the head group [18,19,20]. Differentially citrullinated LL-37 comprising of one to five modified arginine residues (net charge +5 to +1) and wildtype LL-37 showed a similar α-helical structure in zwitterionic detergents as well as lipids 1-Palmitoyl-2-oleoyl-sn-glycero-3-phosphoethanolamine (POPE) and 1-Palmitoyl-2-oleoyl-sn-glycero-3-phosphocholine (POPC) [37]. The citrullinated derivatives, with their decreased number of positive charges, showed a smaller affinity towards LPS and negatively charged phospholipids indicating a decreased interaction with the lipid head group [37,38].

The sequence-specific assignment of functions in LL-37 is of great interest for the rational design of mutants and truncations with enhanced activity. Previously, a truncation-mutation combined approach led to a 24- residue peptide, OP-145, which is in phase II clinical trials [3,39], but new fragments may be constructed on the current basis of information together with future structures. Nowadays, the source of our knowledge is the nuclear magnetic resonance (NMR) structures, co-crystal structures of LL-37 in the presence of detergents, and the selective biophysical and activity-based analysis of mutants and truncations. For example, from truncation studies, it is known that the first four residues are not essential for antimicrobial activity but are involved in the oligomerization of the peptide [25]. This was corroborated by various constructs containing only the LL-37 core region, which maintained the activity [40]. Structure changes of the F6W and F17W induced by carbonate compared to pure water indicated that folding of the peptide and oligomerization hid these residues in a hydrophobic environment. The same mutants, when investigated in the presence of lipids, experienced an even stronger hydrophobic environment and the Lys10 residue coming close to Phe6 [41]. Studies of LL-37 in the presence of detergents using NMR and X-ray crystallography lead to the discovery of Arg23 (but not Arg19 and Arg29) and all Phe residues (Phe5, Phe6, Phe17, and Phe27) as important determinants for molecular interactions, in particular with the major bacterial lipid phosphatidylglycerol (PG) or the detergents used for structure determination, respectively [19,20,42]. 

## 2. The Monomeric Structures of Full-Length and Truncated LL-37

Early NMR investigations of three LL-37 fragments LL-12 (Leu1-Lys12), FK-13 (Phe17-Arg29), and IG-25 (Ile13-Ser37) in sodium dodecyl sulfate (SDS) micelles lead to the determination of flexible substructures [40]. All are α-helical with partially disordered termini, and the FK-13 truncation was defined as the core micelle interacting structure based on TOCSY trimming with significant remaining antimicrobial activity (Figure 1B). Another partial structure termed LL-23 (Leu1-Arg23) was determined in DPC micelles and is also α-helical with the biologically active Ser9 residue dissecting the hydrophobic phase (Figure 1C) [43]. The structures of the truncated GF-17 (Phe17-Val32) and GE-18 (Glu16-Val32) (both with a glycine added to their N terminus) in SDS micelles are also α-helical and were used to demonstrate that the Arg23 residue was important in antimicrobial killing (Figure 1C) [44]. 

In 2008 two LL-37 full-length structures in the presence of SDS and dioctanoyl phosphatidylglycerol (D8PG) (identical structures) were determined by Wang’s group, while a structure in DPC was determined by the Ramamoorthy lab [18,19]. The structure determined in SDS micelles shows a strongly bent helix-break-helix structure (bent at Gly14-Glu16) with a flexible N- and C-terminus, and it was argued that the curvature of the helix was imposed by the micelle topology (Figure 1D,G) [18]. By contrast, the structure determined in DPC displays a different helix-break-helix conformation (break at Lys12) with the termini solvent-exposed and disordered (Figure 1E,G) [19]. Here, the labeling of detergents allowed to generate more detailed structural information and suggested a direct interaction between all four phenylalanine residues and the hydrophobic core of the micelle. Another structural feature determined by Porcelli et al. was the intramolecular salt bridge between Lys12 and Glu16 which is believed to be important for folding and stabilization. Although the two LL-37 full-length structures show significant deviations when superimposed, they both imply the importance of the phenylalanine residues’ interaction with the detergents.

LL-37 was also crystallized using DPC micelles in similar NMR conditions as above and also yielded a monomeric structure in the presence of 70% 2-methyl-2,4-pentanediol (MPD) [20]. This monomeric structure shows a straight α-helix which aligns with the NMR structures in the C-terminal part (residues Ile13 to Ser37) but does not show the kink at Ser12 and significantly deviates here (Figure 1F,G). Various salt bridges stabilize this structure intramolecularly (Asp4/Arg7, Glu16/Arg19, Gln22/Asp26, Asp26/Arg29), but the salt bridge between Lys12 and Glu16 suggested by the NMR data was not observed. Only recently, another monomeric atomic structure of the core fragment LL-37 (Phe17-Arg29) was published and shows the fiber-like supramolecular organization of the peptide in the crystal packing, which was also confirmed by electron microscopy (Figure 1H and next section) [45]. 

## 3. The Dimeric Structure Shows Strong Conformational Plasticity

Until recently, the main source of structural information on LL-37 were monomeric NMR structures determined in the presence of detergents, but the oligomeric state of LL-37 was previously determined to be dimeric-hexameric [20,24,25]. In order to solve this contradiction, we crystallized the peptide in the absence and presence of detergent molecules [20]. The structure of LL-37 in the absence of detergents shows an antiparallel peptide dimer of 5 nm length, primarily stabilized by hydrophobic interactions and H-bonds between Glu16 (monomer 1) and Ser9´ or Lys12´ (monomer 2) at the dimer interface as the main contribution for stability (Figure 2A,B). Hydrophobic residues dominate one of the two faces of the brick-shaped molecules interacting with the hydrophobic portion of membranes. The opposite surface patch of the dimer is continuously hydrophilic and displays a strong surplus of 12 positive charges (Figure 2A).

In order to achieve a membrane-mimicking environment for LL-37 and to investigate this activated membrane state, we determined the structure in the presence of LDAO (PDB-entry: 5NNK) and DPC (PDB-entry: 5NNT). The structures are identical, but the LDAO structure showed three detergents with DPC occupying only the terminal detergent position (equivalent to LDAO1). Both structures showed an unexpected structural transition (5NNK; Figure 2C) [20]. In particular, at the N-terminus, significant structural plasticity of residues Leu1-Arg8 switching from a helical to a random coil state, consequently leading to a structural rearrangement of 2 nm from the corresponding Leu1 residues, is observed. Similar transitions at the C-terminus accompany these changes together with side-chain movements. The changes at the N-terminus realign two phenylalanine side chains (Phe5 and Phe6) from the dimer interface to a surface-exposed scaffold to create interaction sites with the alkyl chains of LDAO. Three of such detergent binding sites (termed L1–L3 in Figure 2C) per monomer have been observed, two at the center and one at the N-terminus with a vertical orientation of the detergents in clusters C1 and C2. The localization and architecture of these sites suggest two lipid molecules at the hydrophobic core of the peptide dimer (C2—see Figure 2) and one at the N-terminus interacting with a second L1 at the N-terminus in the fiber structure (termed C1) (Figure 2E). The distance of the alkyl groups for C2 is similar to alkyl chains in standard lipids, while the distance obtained for C1 is larger. Recently a tetrameric channel structure of LL-37 has also been obtained in the presence of DPC (Figure 2D) [21]. This channel structure is a distorted dimer of dimers, but the architecture of the dimer is different from 5NNK. Interestingly, when integrated in planar lipid membranes, the channel displays a defined conductivity, although the structure of the channel is rather narrow and requests additional movements of the side chains to open for ions.

## 4. Fiber-Like States of Full-Length LL-37 and Truncated Version LL-37_12–29_

The formation of fiber-like structures by antimicrobial peptides has been reported to be stimulated by the presence of lipid vesicles or detergents. Among the first studies, the fluorescently labeled cationic peptide LAH4 and the antimicrobial peptide BTD-2 were shown in fiber states [46,47]. The first LL-37 fiber structures were reported in the presence of lipid vesicles using saturated or unsaturated lipids studied by atomic force and electron microscopy with a diameter of 10 nm [48]. Later, the LL-37 fiber formation in the presence of the detergents LDAO or DPC was confirmed by X-ray crystallography and demonstrated a fiber diameter of 5 nm (Figure 2E) [20]. In this fiber-like structure at atomic resolution, the detergent molecules act as a hydrophobic kit that connects the dimeric LL-37 at the termini (C1 sites) and towards the second layer of peptides (C2 sites). The repetition of this unit results in the formation of the zigzag fiber-like structure containing a shielded hydrophobic core and detergent hot spots (C1/C2) along the fiber axis. The in vivo existence of the LL-37 fibers observed in the crystal lattices was confirmed by labeling the C-terminus with nanogold particles and incubating the peptide with DOPC:DOPG (3:7) small unilamellar vesicles which were studied by cryo-electron microscopy [20]. Recently, the LL-37 active core peptide (residues 17–29) demonstrated to be able to self-assemble into a protein fibril of densely packed helices (Figure 1H) [45]. The crystal revealed that the wide fibrils were composed of amphipathic helices self-assembled into a densely packed and elongated hexameric structure, forming a central pore. This fibril shows an alternance between hydrophobic and cationic zigzagged belts that allow the interaction with the negatively charged bacterial membranes. The fibrillar assembly of this truncation is due to the association of four-helix bundles, each stabilized by a closely packed hydrophobic core. This net resembles the phenol soluble modulin α3 (PSMα3) cross-α amyloid fibrils composed entirely of α-helices perpendicularly stacked to the fibril axis into mated “sheets”, as the β- strands assemble in amyloid cross-β fibrils [49].

## 5. LL-37 Targets Sites in Lipids and LPS

Sequence-specific functional information based on the structures of LL-37 and target molecules such as proteins, DNA, or RNA cannot be deduced due to the missing co-crystal structures. However, there is a clear indication that some of the conserved interactions in detergents likely resemble lipids or LPS binding sites. While interactions between antimicrobial peptides and lipids are more difficult to study at high resolution, detergents can be used to estimate their putative structural influence on AMPs. Similar to membrane proteins, most LL-37 structures are observed in the presence of detergents which can significantly change their secondary, tertiary, and quaternary structure. In the NMR and X-ray structures of LL-37 bound to DPC, localized detergent binding sites involving all phenylalanines were discovered [19,20]. In these structures, lipid interactions were determined by biophysical studies. 

Such detergent binding sites in membrane protein structures are often conserved and resemble the binding sites typically occupied by lipids in vivo (Figure 1A). The aromatic residues observed in the vicinity of detergents or lipids are stacking with their head groups towards the detergent head groups. Here, the alkyl chains of detergents next to each other in space show an orientation resembling natural membrane lipids [50]. Interestingly, the sites for LDAO in our X-ray structure were induced through the presence of the detergent. This fact entailed the structural remodeling of the N-terminus of LL-37 and affected the conformation of the aromatic side chains of Phe5, Phe6, and Phe27 that became able to accommodate the alkyl chain [20]. 

Positive charges in antimicrobial peptides are associated with interactions to the negatively charged cell envelope structures LPS and LTA [6,7,8]. For FhuA and more LPS-interacting proteins, the binding modes have been analyzed in various crystal structures, and specific positively charged residues were determined to be primarily forming H-bonds with the phosphate head group moieties of LPS in a distance of around 13 Å [51]. In FhuA, phenylalanine residues contribute to the binding by forming Van der Waals contacts with the alkyl chains of LPS, an arrangement reminiscent of LL-37 [51]. LPS binding of the antimicrobial peptide pardaxin was also demonstrated to include three phenylalanine residues and, more importantly, two lysine residues arranged at a distance of 12 Å in agreement with the LL-37 scaffold [52]. Moreover, in MSI-594, it was also demonstrated that positively charged residues at a distance of 14 Å are in an optimal arrangement to bind to the negatively charged phosphates in LPS [53]. We propose that based on the structure determined in detergents, LL-37 may form similar LPS binding sites based on residues pairs Arg23/Lys12 (17 Å) and Arg19/Lys12 (13 Å) or Arg19/Lys8 (13 Å), all of which are conserved residues (see Figure 1A). 

## 6. Perspective

Crystal and NMR structures of LL-37 together with biophysical data collected in the absence and presence of detergents, lipids, LPS, and LTA point towards a highly flexible AMP molecule observed in various oligomeric states. Structures determined in the presence of detergents reveal molecular interactions suggesting lipids and LPS as targets of LL-37 in vivo. While lipid targeting of proteins and peptides is only scarcely described, it is a new hallmark of LL-37. Similarly, conclusive studies on the structural plasticity of any other AMP were not performed yet. However, more co-crystal structures with LPS, LTA, or proteins such as PhoQ will shine a light onto the molecular interfaces between LL-37 and other bacterial target molecules, allowing the design, for example, of a peptide optimized to maintain antimicrobial activity but devoid of PhoQ interactions. Moreover, rationally or structure-based designed point mutants or truncations of LL-37 can yield derivatives with an increased target affinity, for example, towards LPS or lipid membranes. A combination of both modifications using a random sequence screening was used to generate the OP-145 peptide with a high MIC but a lower affinity for red blood cells and properties in the treatment for middle ear infections (phase II clinical trials) [54]. While all LL-37 mutants so far were designed on random screens due to missing co-crystals structures, real complexes are the start of a rational drug design like in the proline-rich peptides bound to the *Thermus thermophilus* 70S ribosome [55]. Furthermore, LL-37 in complex with detergents as synergistic molecules suggest they can also be used for medical applications. Finally, recent interaction studies demonstrate valuable targets of LL-37, including the spike protein of SARS-CoV-2, and here for the first time, co-crystal structures of this complex could reveal the mechanism by which LL-37 suppresses virus entry (Roth A. et al., 2020 currently in BioRxiv).

## Figures and Tables

**Figure 1 ijms-22-05200-f001:**
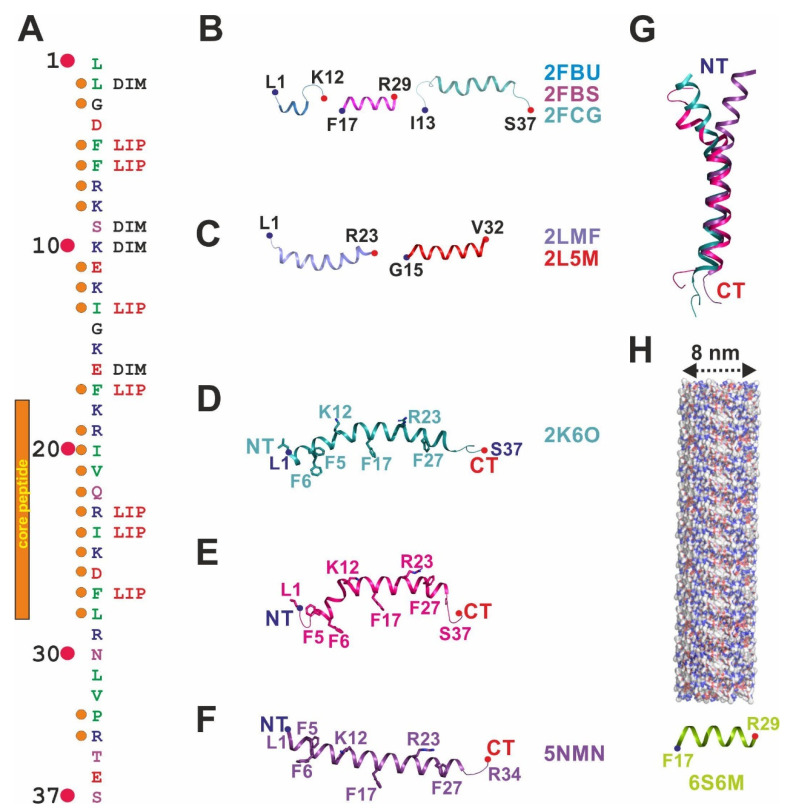
Monomeric full-length and truncated structures of LL-37 determined in the presence of detergents are all α-helical. (**A**) Sequence of LL-37 to visualize hydrophobic residues color-coded in green, positively charged residues in blue, negatively charged residues in red, and hydrophilic uncharged residues in magenta. Combined evolutionary conservation and structure-based functional assignment are given. Orange dots on the left side of the sequence (marked by numbers 1–37) indicate high sequence identity, and structure-based functions for residues are assigned on the right (DIM—dimerization interface; LIP—lipid-binding site). (**B**) Structures of LL-12 (Leu1-Lys12), FK-13 (Phe17-Arg29), and IG-25 (Ile13-Ser37) determined in the presence of sodium dodecyl sulfate (SDS) or dioctanoyl phosphatidylglycerol (D8PG) by nuclear magnetic resonance (NMR) in cartoon representation and color-coded in three different colors with N-terminus (NT), and C-terminus (CT) are marked. The first and last residue is given together with the PDB-entry code. (**C**) Subsequently, published structures of LL-37 fragments solved by NMR, including the antimicrobial inactive LL-23 (Leu1-Arg23) mutant derivative, show α-helical structures. (**D**) The full-length structure of LL-37 determined in the presence of SDS by NMR shows a bent two-partite structure (residues implicated in detergent binding are shown in stick representation). (**E**) Full-length structure of LL-37 in dodecylphosphocholine (DPC) determined by NMR displays defined detergent-peptide interactions and a kinked α-helix. (**F**) Crystal structure of LL-37 determined in the presence of DPC shows defined detergent binding sites representing lipid and lipopolysaccharide (LPS) binding sites in vivo, respectively. The structure shows a disordered but structured part at the N-terminus (residues Leu1-Arg7). (**G**) Superposition of the three LL-37 structures (see (**D**–**F**)) shows a significant structural deviation at the N-terminus. (**H**) Structure of the LL-37 (Phe17-Arg29) derivative crystallized in a packing that allowed the construction of fibers.

**Figure 2 ijms-22-05200-f002:**
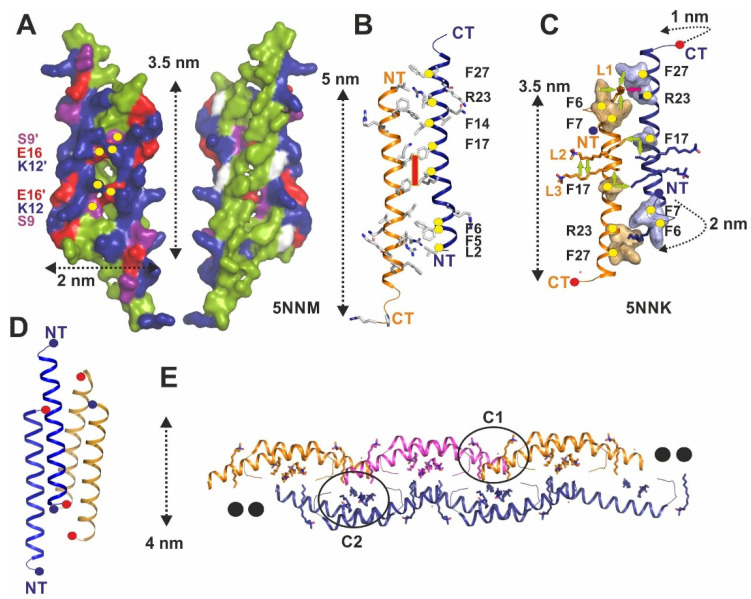
Crystal structures of the dimeric and tetrameric full-length LL-37 show significant structural rearrangement. (**A**) Surface structure representation of LL-37 crystallized in the absence of detergents (PDB-entry: 5NNM). There are two faces of the peptide, one of which is strongly charged (surplus +12/dimer; Arg and Lys are in blue, Asp and Glu in red and Tyr, Ser, Thr, Asn, Gln in magenta), while the opposite side is hydrophobic (all hydrophobic residues are color-coded green). Hydrophilic residues at the interface forming H-bonds are marked by yellow dots. (**B**) Structure of LL-37 is determined in the absence of detergents. The 5 nm long peptide complex is shown in the same orientation as (A) with the N- (NT), the C-termini (CT), and the twofold symmetry axis (red square) marked. Conserved residues at the dimer interface are marked with yellow dots, according to the published analysis [22]. (**C**) Structure of LL-37 is determined in the presence of LDAO. Three detergent molecules (L1-L3) per monomer could be identified in the structure, two of which in the center of the dimer with the alkyl chains oriented perpendicular to the peptide chains and interactions between detergent and LL-37 are marked by green arrows. One detergent was observed between the head-to-tail arranged dimers in the crystal packing (see also (E)). (**D**) Tetrameric channel structure of LL-37 formed by two antiparallel dimers (shown in orange and blue). (**E**) Head-to-tail polymerization and fiber formation (diameter 4 nm) of LL-37 crystallized in the presence of LDAO and DPC. Two LDAO detergent molecules are encapsulated in the pocket, which is formed after polymerization of the dimeric peptide. There are two reoccurring clusters of detergents observed along the fibril, one between the head and tails of the polymer (C1) and another one at the center of the dimers (C2).

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
