# Peer review of "Structural Plasticity of LL-37 Indicates Elaborate Functional Adaptation Mechanisms to Bacterial Target Structures"

_ijms, 2021, doi:10.3390/ijms22105200_

Round 1

Reviewer 1 Report

This is an interesting, well written review. I have some minor remarks:

  1. In the Introduction, there is an information that 36 AMPs are in clinical trials. Are any AMPs already approved for the use in the clinics?
  2. The Authors include the information that OP-145 in in clinical trials.  What are the indications for its use? Is there any data available on the results of Phase I or II trials? Is OP-145 still in clinical trials, since the first information about such trials is from 2006?

Author Response

Dear reviewer 1,

Thanks for your comments and questions regarding our review.

1. In the Introduction, there is information that 36 AMPs are in clinical trials. Are any AMPs already approved for use in the clinics?

As detailed in the recent reviews regarding AMPs in clinical trials (Browne et al., 2020; Chen and Lu, 2020), all AMPs currently tested and approved for medical treatment of defined diseases (e.g. colistin, gramicidin, polymyxin B) belong to the non-ribosomally produced AMPs(AMPs synthesized by non-ribosomal peptide synthetases) (Liu et al., 2019). Nowadays, there are about 40 gene-encoded AMPs (e.g. LL-37, defensins) approved for clinical investigations, but the number of ribosomal AMPs in clinical and preclinical stages is increasing (Koo and Seo, 2019; Browne et al., 2020; Dijksteel et al., 2021).

2. The Authors include the information that OP-145 is in clinical trials.  What are the indications for its use? Is there any data available on the results of Phase I or II trials? Is OP-145 still in clinical trials, since the first information about such trials is from 2006?

Thanks for this question. The lack of information regarding the OP-145 clinical trials is obvious. This LL-37 derivative was proposed to be used in chronic suppurative otitis media (Browne et al., 2020). As the reviewer says, the last release of information regarding these clinical trials was done several years ago by the Octopus company(https://www.biospace.com/article/releases/octoplus-initiates-phase-ii-clinical-trials-with-op-145-in-chronic-middle-ear-infection-and-presents-positive-phase-i-results-/). We have not been able to find an update on the results of phase I or II trials. In fact, we attended a conference some time ago where it was revealed that the OP-145 clinical trials were stopped due to stability issues, although no available information on the internet can confirm it. In our review, we tried to include only the information contained in published papers/databases, and in all of them, OP-145 is still included as an AMP in clinical trials.

References

Browne, K., Chakraborty, S., Chen, R., Willcox, M. D., Black, D. S., Walsh, W. R., et al. (2020). A New Era of Antibiotics: The Clinical Potential of Antimicrobial Peptides. Int. J. Mol. Sci. 21. doi:10.3390/ijms21197047.

Chen, C. H., and Lu, T. K. (2020). Development and Challenges of Antimicrobial Peptides for Therapeutic Applications. Antibiotics (Basel) 9. doi:10.3390/antibiotics9010024.

Liu, Y., Ding, S., Shen, J., and Zhu, K. (2019). Nonribosomal antibacterial peptides that target multidrug-resistant bacteria. Nat. Prod. Rep. 36, 573–592. doi: 10.1039/c8np00031j

Koo, H. B., and Seo, J. (2019). Antimicrobial peptides under clinical investigation. Pept. Sci. 111, 715.

Dijksteel GS, Ulrich MMW, Middelkoop E, Boekema BKHL. (2021). Review: Lessons Learned From Clinical Trials Using Antimicrobial Peptides (AMPs). Front Microbiol. 12:616979. doi: 10.3389/fmicb.2021.616979. 

Reviewer 2 Report

This minireview is quite an extensive analysis of structural data of full-length and truncated forms of human cathelicidin LL-37 in the lite of their interactions with different lipid moieties. This paper discusses the structural plasticity of LL-37 and the implications it has in terms of lipid targets as well as the potential antimicrobial functions. The topic of this review is scientifically sound especially in the context of increasing antibiotic resistance and the search for new or modifying the existing agents with antimicrobial potential. Overall the manuscript is well written.

Author Response

Dear reviewer 2,

Thanks for your comments. 

Best wishes,

The authors